# Qualitative exploration of the constraints on mothers' and pregnant women's ability to turn available services into nutrition benefits in a low-resource urban setting, South Africa

Agnes Erzse ,[1] Chris Desmond,[1,2] Karen Hofman,[1] Mary Barker,[3,4] Nicola Joan Christofides[5]

For numbered affiliations see end of article.

**Correspondence to**
Agnes Erzse;
agnes.erzse@wits.ac.za

## ABSTRACT

**Objectives** Despite free primary healthcare services and social protection system for mothers and children, significant nutrition inequalities occur across the globe, including in South Africa. This study aimed to explore what determines mothers' ability to access and turn available services into nutrition benefits.

**Design** An exploratory qualitative study was conducted including semistructured interviews with employees from community-based organisations and focus groups with pregnant women and mothers. Discussions focused on existing services perceived as important to nutrition, differences in mothers' ability to benefit from these services, and the underlying unmet needs contributing to these disparities. Data were analysed thematically using a novel social needs framework developed for this study where social needs are defined as the requisites that can magnify (if unmet) or reduce (if met) variation in the degree to which individuals can benefit from existing services.

**Setting** A resource-constrained urban township, Soweto in Johannesburg.

**Participants** Thirty mothers of infants (<1 year old) and 21 pregnant women attending 5 primary healthcare facilities participated in 7 focus groups, and 18 interviews were conducted with employees from 10 community-based organisations.

**Results** Mothers identified social needs related to financial planning, personal income stability, appropriate and affordable housing, access to government services, social support and affordable healthier foods. The degree to which these needs were met determined mothers' capabilities to benefit from eight services. These were clinic-based services including nutrition advice and social work support, social grants, food aid, community savings groups, poverty alleviation projects, skills training workshops, formal employment opportunities and crèches/school feeding schemes.

**Conclusion** Findings demonstrate that while current social protection mechanisms and free health services are necessary, they are not sufficient to address nutrition inequalities. Women's social needs must also be met to ensure that services are accessed and used to improve the nutrition of all mothers and their children.

## STRENGTHS AND LIMITATIONS OF THIS STUDY

⇒ Current approaches to understanding and addressing nutrition inequalities are limited; new methods are needed that consider the variation in individuals' ability to benefit from available services.

⇒ This is the first study in South Africa to understand the variation of nutrition outcomes of mothers and children through a social needs approach.

⇒ This paper reports on the development and application of a novel framework anchored in a an ecological and capabilities approach, to help better understand social needs that limit mothers' ability to turn available services into better nutrition outcomes.

⇒ Perspectives of mothers of young children, pregnant women and community-based organisations were included, enhancing both the validity and comprehensiveness of the data.

⇒ Participants' views represented the experiences of lower income mothers in a resource-constrained urban township; there is a need to conduct further studies at different locations, where mothers' social needs may take different forms.

## INTRODUCTION

Poor maternal and child nutrition is one of the biggest preventable causes of ill health and poor human development globally. Despite its commitment to the 2025 global nutrition targets set by the World Health Assembly,[1] South Africa is not on track to achieve a 30% reduction in low birth weight and 40% in stunting. Over the past decade, no progress has been made with low birth weight remaining the same from the benchmark of 14.8% and current prevalence of 15%.[2] Stunting among under 5 years has risen from 23% in 2012, to 27% of children under five being stunted in 2023.[3] Moreover, under five childhood overweight and obesity rates are

essentially unchanged.[4] Similarly, no progress has been made achieving a 50% reduction in anaemia in women from 28.6% in 2012, with current figures being at 30%[5] and 33% of pregnant women are obese or overweight and 9% have gestational diabetes.[6]

There are significant inequalities in nutrition outcomes across and within countries. Disparities are particularly evident in South Africa. Among those at the bottom end of the distribution are mothers and children in lower income urban townships, who have worse nutrition outcomes than the national average. Soweto, for example, the largest township in South Africa, has more low birth weight babies (1.6 percentage points), more stunted and obese children under the age of 5 (7 and 11 percentage points, respectively),[7] and more obese mothers (33 percentage points).[8]

These disparities exist despite free maternal and child healthcare; the presence of high impact first 1000 days interventions,[9] and social protection mechanisms such as the Child Support Grant (South African Rand (ZAR) 450 or US$26/month as of 2023).[10] This suggests that the availability of universal primary care services may not be enough to ensure good nutrition for all mothers and their children. Families may need to access different types and amounts of services and resources such as policies, foods, changes in social norms and infrastructure, to achieve the same level of optimal nutrition. This variation challenges the way we think about prioritising and investing in interventions for nutrition. Both globally and in South Africa, the focus has been on increasing the coverage of individual level curative and population-level preventative nutrition programmes.[11–13] Less consideration has been given to what makes it possible for individuals to benefit from these existing services.

If nutrition is influenced by variations in the needs and opportunities that mothers and pregnant women face, a new theoretical approach that embraces this complexity is necessary. The aim of this paper is to examine these variations through engagement with key stakeholders (mothers, pregnant women and community-based organisations (CBOs)) in Soweto and to operationalise a novel conceptual framework combining ecological and capabilities approaches.

## Existing theoretical approaches: missing social needs?

Bronfenbrenner (1977) was one of the earliest theorists to capture the complexities of human development through his ecological model comprised of an arrangement of nested circular systems.[14] His model posits that an individual's development is the result of the dynamic interplay among multiple systems of influence, including the micro- (innermost circle), meso-, exo- and macro- (outermost circle) system. A fundamental principle that sets the microsystem and mesosystem apart from the exosystem and macrosystems is the extent to which these reflect individual experience. Bronfenbrenner envisioned that the microsystem and mesosystems refer to the face-to-face interaction of individuals with the immediate settings such as home, neighbourhood and social networks. By contrast, the exosystem and macrosystem consist of social structures (e.g., institutions like government agencies) that regulate individuals' lives at the meso level and micro level yet are removed from the individual.[15] Bronfenbrenner's micro level, meso level and macro level have become the foundation for common typologies in the public health literature to describe the drivers of, and strategies to improve nutrition.[16] Table 1 summarises some of the most used typologies in the nutrition literature as they relate to the ecological levels.

At the micro level, immediate determinants of nutrition (e.g., iron deficiency) are commonly addressed through medical, nutrition specific actions (e.g., iron supplementation).[17] Using the metaphor of a stream,[18] these actions can be described as downstream. By contrast, upstream actions are located at the macro level and are viewed as nutrition sensitive[17] as they address the underlying structural and social determinants of health, such as poverty and unemployment. Interventions addressing social determinants of health range from education policies, taxes on unhealthy foods to interventions to improve women's status in society,[17] none of which involve the individual directly.

Notwithstanding the need for services at the micro level and macro level to improve nutrition, the inequalities described earlier suggest a need to identify meso level strategies that address sources of variation in individuals' capability to benefit from existing services. Sources of variation could include personal (e.g., possession of identity documents) and socioeconomic factors (e.g., cost of healthy food), that can lead to avoidable variation in capability to benefit from services if the associated need—referred to as social need—is not met. In this context, social needs are the requisites that can magnify (if unmet) or reduce (if met) the sources of variation individuals face when benefiting existing services. As such, variations

| Table 1 | Overview of common typologies used to describe drivers of and interventions for nutrition with examples | | |
|---|---|---|---|
| **Ecological model** | **Determinants** | **Streams** | **Specificity** |
| Micro level | Nutrition needs (e.g., iron deficiency) | Downstream (e.g., iron supplementation) | Nutrition specific |
| Meso level | Social needs (e.g., transportation, housing) | Mid-stream (e.g., community-based services) | Nutrition sensitive |
| Macro level | Structural/social determinants of health (e.g., poverty and social exclusion) | Upstream (e.g., social safety nets) | |

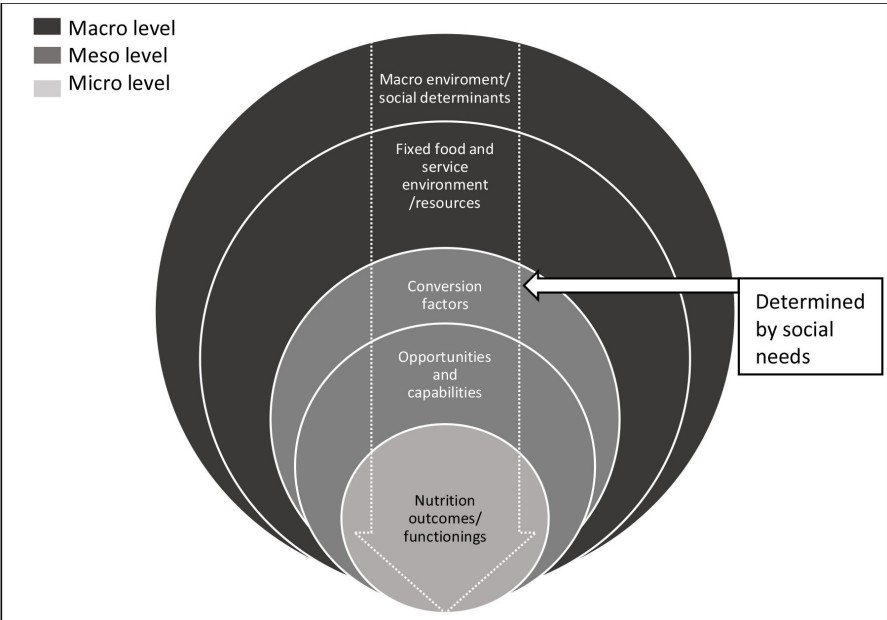

**Figure 1** A social needs framework combining the capabilities approach[19] and the ecological model.[52]

in social needs influence the effects of both macro level and micro level services on nutrition and may therefore be responsible for consequent disparities in nutrition outcomes.

There is a clear overlap between social needs and 'conversion factors', a concept central to Amartya Sen's 'capability approach'.[19] The capability approach focuses on the conversion of resources into individual capabilities (real opportunities) to achieve a valuable combination of human functionings, termed 'beings' and 'doings'. From a nutrition point of view, resources can be services and 'beings' and 'doings' can include being well nourished. The capabilities approach recognises that there are personal, social and environmental aspects of the environment (conversion factors) that either expand or restrict one's real opportunities (capabilities) to convert resources into valued functionings.[20] Similarly, social needs play key roles in determining the degree to which mothers can benefit from government and healthcare services for optimal nutrition. The theoretical complementarity between the ecological model and the capabilities approach provides reason to believe that the integration of the two could help better understand what mothers in their communities need so that they are capable of converting available government and healthcare services into better nutrition outcomes.

### A social needs framework

The two models combined (figure 1) allow for mothers' diversity of needs and capabilities to benefit from existing services. Furthermore, it reflects the interaction between ecological levels by demonstrating how such diversity at the meso level determines the effect of an existing service environment on nutrition.

The framework groups all services into a combined service environment at the macro level, based on the assumption that a 'service is a service' for an individual regardless of whether the service targets downstream or upstream determinants. The model seeks to highlight is that the availability of a service is not an end in itself. The effect of the service (indicated with the arrow on figure 1) is determined by the conversion factor as it passes through the meso level that determines mothers' real opportunities (capabilities) to benefit from services. This reflects variation in mothers' experiences of a given service environment and the degree to which they can convert services into nutrition benefits. One might fully benefit from a service by accessing and implementing it, another might partially benefit by accessing it but not being able to use the benefits it offers, and a third might not be able to access or benefit from the service. Where an individual sits on this benefit scale is determined by their social needs and the degree to which these social needs are met. In this sense, the framework identifies conversion factors as the product of social needs.

Not all services will see variation in mothers' capability to benefit (conversion factors), and not all variation will be influenced by the same social needs. By applying this framework, this article aims to explore which services (either macro or micro) see big variation in mothers' conversion factors (capability to benefit) and identify some of the social needs that shape these conversion factors. This framework is particularly relevant in dealing with the nutrition of disadvantaged people or of socioeconomic groups in unfavourable circumstances or conditions because they are likely to have multiple unmet social needs.

One approach to increasing understanding of social needs is to engage those who have direct experience of these needs. These include mothers and pregnant women, as well as CBOs, defined as any non-profit organisation

operating at meso levels in community settings. The rationale behind engaging CBOs is that they interact and monitor community members and are therefore able to identify their unique challenges.

In a context of available macro level government and micro level nutrition services in Soweto, Johannesburg, this research sought to understand (1) existing services which are perceived as valuable for nutrition but that not all mothers can benefit from equally, (2) the sources of variation in mothers' capability to benefit from those services and (3) associated social needs that if met could reduce this variation.

## METHODS
This cross-sectional, exploratory qualitative study was conducted in Soweto between February and October 2022. It involved semistructured individual interviews (interviews) with employees from CBOs and focus group discussions (FGDs) with adult (>18 years) pregnant women and mothers of infants under 1 year of age attending antenatal and postnatal clinics.

### Setting
Soweto is a densely populated working class township which by 2019 contained about 1.8 million people, one-third of Johannesburg's population.[21] Inhabitants are almost entirely black South Africans and 80% of adults are unmarried.[22] Forty per cent of households are headed by women and one in two children grow up without their father in the home.[23] In some parts of Soweto, individuals endure household and community violence, crime and extreme poverty, and 19% of all households receive no income other than state grants (if eligible).[22] Lower income areas are characterised by unpaved roads, lack of drainage systems, houses with tin roofs, shared drinking water sources and shared toilet facilities.[24 25] Coresidence of several families within a four roomed government house is a common feature, with many accommodated in smaller rooms or shacks built at the back of the house.[24] A 2018 survey found that young mothers in Soweto are particularly vulnerable to poor-quality housing, experience lower levels of education and high rates of unemployment (64%).[25] Compared with nulliparous women, young mothers are also more likely to face food insecurity and have poorer health outcomes, including higher levels of overweight and obesity.[25]

### Sampling
The selection of interview participants began by developing a contact list of Soweto-based CBOs from an internet search, including two CBO directories.[26 27] CBOs were identified as eligible if the mission/vision statement and overview of services contained explicit language around any of the following: provision of maternal and child health services, social support and nutrition to children, mothers and pregnant women. CBOs focusing primarily on older persons, people living with chronic infection (e.g., HIV and AIDs, TB), disabled people, crime prevention and drug abuse were excluded. Out of the 15 CBOs that were identified and contacted, 10 responded to the invitation to participate, thus comprising our final sample. An invitation letter was sent to CBOs, followed up with a telephone call. Potential participants put forth by the CBOs were approached by the first author and asked if they were willing to participate in the study.

FGD participants included purposively sampled pregnant women attending antenatal care and mothers with infants under 1 year of age attending postnatal care at five primary healthcare facilities in Soweto (online supplemental file 1). Clinics were purposefully selected to ensure representation from various suburbs, allowing for a more comprehensive collection of perspectives and experiences. Recruitment was undertaken by a local multilingual qualitative researcher with experience of working with mothers in Soweto. The researcher approached individuals in communal areas of the healthcare facilities (e.g., the main corridor, waiting areas) and invited those interested if they resided in Soweto, they were pregnant or had at least one infant under 1 year of age, and were 18 years of age and above. The sample size for both the interviews and FGDs was determined by data saturation.[28]

### Data collection
An interview and a FGD guide were developed separately to assist data collection (see online supplemental files 2 and 3). Guides asked participants about their perceptions of the challenges that affected maternal and child nutrition in the community, as well as the available resources or services that they knew of and might help addressing these challenges. The first author, who had experience in qualitative interviewing and working with communities in Soweto, conducted the interviews in English, either in person at the CBO or via telephone. The FGDs were facilitated by a trained, multilingual researcher, familiar with customs and traditions of the community. FGD participants expressed themselves in English and in their vernacular languages (Xhosa, Zulu, or Sotho). FGDs were organised after the completion of the interviews, and these took place at a research centre in Soweto. All FGD participants completed a sociodemographic questionnaire. A pilot to test the data collection methods was undertaken with participants who met the eligibility criteria. The pilot was conducted in the study setting. Results from the pilots were included in the analysis. The interviews and FGDs lasted about an hour and all participants were reimbursed for time and transport. Written consent was taken from all participants.

### Patient and public involvement
The data collection followed an iterative process, whereby interviews with CBOs informed the development of the FGD guide. Further, early FGDs resulted in adjustments to subsequent ones to enhance the relevance of the questions. A summary of the findings was presented for

validation to participants and patients (attending antenatal and postnatal care).

## Data management and analysis

Interviews and FGDs were audio recorded, transcribed and translated into English where necessary. Transcripts were analysed progressively to determine the point at which saturation was achieved. The first author used a hybrid (deductive and inductive) approach to build the coding structure and coded the interviews and FGD transcripts with MAXQDA 2022 data analysis software[29] using thematic analysis.[30] Data were triangulated from the interviews and FGDs to identify similarities and differences and looked at convergence patterns to validate interpretation. Once inductive and deductive codes were developed, codes were organised into broader categories based on thematic similarities. Lastly, categories were reorganised by three deductive themes anchored in our study objectives and framework (figure 1): services valuable for nutrition (but which not all mothers could benefit from), sources of variation and social needs (see online supplemental file 4). The coauthors discussed their interpretations of the data. Verbatim quotes from participants have been selected to illustrate how FGD participants and CBOs accounts were linked to themes. Reporting of the findings adheres to Consolidated criteria for Reporting Qualitative research guidelines[31] (see online supplemental file 5).

## RESULTS

We present the characteristics of participants then discuss the six social needs identified by the participants as they link to multiple sources of variation in mothers' and pregnant women's (hereinafter mothers) capability to benefit from eight different services that both CBOs and mothers identified as valuable for optimal maternal and child nutrition (hereinafter nutrition). Where results are similar for CBOs and mothers, these are presented without participant specification. Where results differ, these are noted and discussed.

## Characteristics of CBO participants

We interviewed 18 employees representing 10 CBOs that serviced clients in 11 of 40 Soweto suburbs. Eight CBOs aimed to directly improve the well-being of orphans and vulnerable children and adolescents (0–18 years of age), and two CBOs aimed to directly improve the well-being of women, including mothers and pregnant women. On average, there were two participants per CBO. All but one participant were women, and they represented Founders and Directors (n=4), managers responsible for directing programmes and employees (n=2), social workers (n=11) and a volunteer (n=1) who worked directly with beneficiaries. Thirteen of the participants indicated that they had been employed at their respective CBOs for over four years.

## Characteristics of FGD participants

Fifty-one women participated in seven FGDs (see group compositions in online supplemental file 1). Twenty-one (41%) were pregnant (12 of them for the first time) and of the 39 non-first-time mothers (71%) had at least two children. Twenty-nine (56%) participants were single while 17 (41%) were partnered or married. All but two participants were unemployed, and the majority relied on social grants (64%) as their main source of income. Thirty-five participants (68%) reported their household income to be less than ZAR 3000 (US$187) per month, 77% being less than ZAR 1000 (US$57.94). Mothers' living conditions differed. Some lived in shacks/informal dwellings (31%), others in owner-occupied houses (bond house) (21%), government subsidised houses (21%), or single rooms built in the backyard of other households (19%). Thirty-two mothers (62%) had stable electricity, 38 (74%) had a fridge, and 10 (19%) had daily access to internet. On average, there were five people living in their houses (range 2–11). Less than half (45%) reported being members of community groups, and 10 (43%) of those who did were part of *stokvels* (community money savings groups). Details of FGD participants are provided in table 2.

## Social needs of mothers

The analysis identified six social needs: financial planning; personal income stability; appropriate and affordable housing; access to civic services, welfare and the labour market; social support; and affordable healthier foods. The following section reports on each of these needs without implying a priority rank order, as they link to multiple sources of variation that enabled or disabled mothers from benefitting from eight different services. These services were recognised by CBOs and mothers as valuable for nutrition but there was variation in mothers' capability to benefit from them. These included clinic-based services (nutrition advice, social workers); social grants (Child Support Grant, Social Distress Relief); food aid including parcels and vouchers (distributed by the Department of Social Development, churches, CBOs, crèches/schools and clinics); stokvels; poverty alleviation projects (sewing, baking groups and food gardens at schools, and CBOs); skills training workshops; formal employment opportunities including learnerships; and crèches/school feeding schemes.

Table 3 categorises our findings according to the concepts of the framework developed for this study (figure 1): social needs, sources of variation and services.

### Financial planning for nutrition

Participants saw nutrition and unmet social needs for financial planning as deeply intertwined. Mothers' ability to save and/or to maintain and multiply scarce nutrition budgets was perceived to determine the impact of five services on nutrition. First, the degree to which a mother could 'stretch' scarce resources including social grants and food vouchers was partly determined by her

**Table 2** Sociodemographic characteristics of FGD participants

| Age (years) | | |
|---|---|---|
| Minimum–maximum; mean (SD) | 18–49 | 31 (8) |
| **Pregnancy status** | N | % |
| Pregnant | 21 | 41.1 |
| Non-pregnant | 30 | 58.8 |
| **Number of babies/children cared for** | | |
| 0—first time mother | 12 | 23.5 |
| 1 | 11 | 21.5 |
| 2–4 | 25 | 49.0 |
| 5 and above | 3 | 5.8 |
| **Marital status** | | |
| Single never married | 29 | 56.8 |
| Married | 4 | 7.8 |
| Partnered | 17 | 33.3 |
| Separated | 1 | 1.9 |
| **Highest level of schooling** | | |
| Primary school | 2 | 3.9 |
| Some high school | 20 | 39.2 |
| Completed high school | 25 | 49.0 |
| Diploma/higher diploma | 4 | 7.8 |
| **Household income** | | |
| ZAR 1000 or less | 27 | 52.9 |
| ZAR 1001–3000 | 8 | 15.6 |
| ZAR 3001–R5000 | 10 | 19.6 |
| ZAR 5001–R10000 | 3 | 1.9 |
| ZAR 10 001–20 000 | 3 | 5.8 |
| **Source of income** | | |
| Government grants | 33 | 64.7 |
| Formal employment | 2 | 3.9 |
| Informal employment | 2 | 3.9 |
| Others* in the household | 8 | 15.6 |
| **Type of house** | | |
| Bond house | 11 | 21.5 |
| Government subsidised houses | 11 | 21.5 |
| Shack or informal dwelling | 16 | 31.3 |
| Single outside room | 10 | 19.6 |
| Other | 2 | 3.9 |
| **Number of people in the house** | | |
| Minimum–maximum; mean (SD) | 1–11 | 5 (2) |
| **Household assets** | | |
| Stable electricity | 32 | 62.7 |
| Radio | 33 | 64.7 |
| A fridge | 38 | 74.5 |
| Television | 38 | 74.5 |
| Cell phone | 33 | 64.7 |

Continued

**Table 2** Continued

| | | |
|---|---|---|
| Daily access to internet | 10 | 19.6 |
| Car | 4 | 7.8 |
| Clothing sufficient to keep mothers and children warm | 38 | 74.5 |
| **Membership** | | |
| Stokvel | 10 | 19.6 |
| Community garden group | 2 | 3.9 |
| Mothers support group | 2 | 3.9 |
| Women's association | 2 | 3.9 |
| Community service (school development committee) | 1 | 1.9 |
| Religious groups | 3 | 5.8 |
| Any | 23 | 45.0 |

*Infants'/children's father, grandparents, other adult cohabitants.
FGD, focus group discussion; SD, standard deviation.

ability to create and maintain a household budget. A CBO participant emphasised that: "If we give them food vouchers, we must make sure that they're able to stretch their food. That is why budgeting is important" *(CBO 5, women, Founder and Director)*. Without financial skills, mothers were perceived to "spend a long time without cash between paycheques. By the time they reach month end, the food is long finished" (*mother of young infant, FGD 2*).

Second, well-planned finances allowed some mothers to save for start-up costs, necessary to fund their poverty alleviation projects. Savings allowed mothers to buy seedlings and tools for gardening projects, or ingredients for a baking group. A CBO participant claimed that mothers without financial skills struggle to plan, "if they've got ZAR 100 [USD 6], instead of buying seeds to invest for the future, they will rather think for today" *(CBO 8, women, project manager)*.

Participants also reported that mothers with financial know-how used social grants and food vouchers in a manner that was more beneficial for nutrition. This is because financially literate mothers spent more effectively on nutrition, purchased in bulk, looked for discounts, and cut expenses by preparing food at home instead of purchasing ready-made baby foods at high costs. Resultant savings were perceived to be used to diversify diets.

Not all financially literate mothers benefitted from services to the same degree. Participants acknowledged that transforming small amounts of money into savings was harder for those with low and uncertain incomes. Services with long-term return on investments such as skills training workshops, or those that required sustained monetary contribution such as stokvels, saw larger variation in mothers' capability to benefit.

**Table 3** Overview of study findings according to concepts of the framework described in figure 1

| Social needs | Sources of variation | Services |
|---|---|---|
| Financial planning for nutrition | ▶ Ability to save, maintain and multiply budget<br>▶ Ability to trade off present investment for future return<br>▶ Know-how of eating healthy on a budget (e.g., buying in bulk, home preparation) | ▶ Social grants<br>▶ Stokvel<br>▶ Poverty alleviation projects<br>▶ Skills training<br>▶ Food aid (vouchers) |
| Personal income stability | ▶ Single motherhood/absent fathers<br>▶ Grant, kinship, loan dependency<br>▶ Lack of control over nutrition in the household | ▶ Clinic-based service (nutrition advice)<br>▶ Social grants<br>▶ Stokvel<br>▶ Crèches/school feeding scheme |
| Appropriate and affordable housing | ▶ Monthly rent payments<br>▶ Overcrowding, sharing of scarce resources<br>▶ Household's food budget must cater for all vs individuals' nutritional needs<br>▶ Poor living conditions (e.g., lack of stable electricity, storage space, refrigeration) | ▶ Social grants<br>▶ Food aid<br>▶ Clinic-based service (nutrition advice)<br>▶ Stokvel |
| Access to civic services, welfare and labour market | ▶ Service eligibility (e.g., possession of identification documents, birth certificate, matric certificate)<br>▶ School dropouts<br>▶ Life skills (e.g., internet literacy, CV writing, form completion)<br>▶ Ability to pay for transportation, printing, internet | ▶ Social grants<br>▶ Food aid<br>▶ Formal employment<br>▶ Skills training |
| Social support<br>*Psychosocial* | ▶ Healthcare and welfare stigma<br>▶ Low self-esteem and hopelessness<br>▶ Poor coping (mental health and substance abuse)<br>▶ Poor partner and family dynamics<br>▶ Domestic violence | ▶ Clinic-based services (antenatal care, social workers)<br>▶ Food aid<br>▶ Social grants |
| *Social cohesion and connectedness* | ▶ Perceived and experienced mistrust, corruption, exclusion<br>▶ Lack of belonging, isolation<br>▶ Social connectedness (e.g., group membership) | ▶ Food aid<br>▶ Poverty alleviation projects<br>▶ Skills training<br>▶ Crèches/school feeding scheme |
| Affordable healthier foods | ▶ Delayed/periodic access to healthier food<br>▶ Ability to pay for transport to supermarkets<br>▶ Awareness of healthier diet (beyond fruits and vegetables that is, legumes, brown rice)<br>▶ Influence of local food environment | ▶ Clinic-based service (nutrition advice)<br>▶ Social grants |

## Personal income stability

The effect of available services on nutrition was seen to be dependent on whether "one has another form of cash except for the social grant" *(mother of young infant, FGD 5)*. For example, clinic-based nutrition advice was seen valuable but its potential effect diminished when mothers lacked personal income stability, as one CBO participant explained: "if you are saying to a mother who has just given birth, they must go get fruits, vegetables. They don't have income for that. In fact, they are waiting for the first grant money" *(CBO 6, women, project manager)*. Moreover, without personal income stability, some mothers lived in overcrowded households, where coresidence compromised mothers' decision-making power over nutrition and restricted their capability to implement any nutrition advice. A CBO participant explained that as a mother who is dependent on others in the household, "you must take whatever is readily available for you. Even if you know that eating maize every day is not good. [...] Maybe today you wanted to say 'can you try fish' - you don't have that voice because you're not financially contributing" *(CBO 1, women, service provider)*.

Other service benefits that were contingent on mothers' personal income stability, included those that required sustained financial contribution. One such service was

crèches with nutritious meals. Mothers recognised these as valuable for children to be well nourished, however not every mother and child could benefit from them. A participant suggested that as a grant dependent mother "You can't take them [children] to a crèche because the social grant money has to buy food" *(pregnant woman, FGD 4)*.

Similarly, participation in stokvels required monthly contributions that only mothers with adequate income stability could afford and benefit from. Participants highlighted that a mother without financial stability "will tell you that they [stokvels] will make her cash short for the rest of the month" *(mother of young infant, FGD 6)*. By contrast, those who were able to benefit from stokvels were seen to be more food secure. This was "because come January until around March you still have food. And if there a few of you in the house, say about 4 of you—it can easily last until July" *(pregnant woman, FGD 4)*.

## Affordable and appropriate housing

Mothers with unmet needs for affordable and appropriate housing were perceived to benefit less from the potential effect of four services on nutrition. First, the degree to which a mother could convert social grants to improved nutrition was dependent on whether she had to pay rent. In the absence of personal income or father's financial support, cost-burdened mothers relied on the

Child Support Grant to cover all household expenses including rent. Participants recognised this as a key source of variation in mothers' capability to purchase the foods recommended at the clinic: "After pay day, one can maybe buy those [recommended] fruits for the first and second week. But how much will one have left from the grant money after paying the rent. These are the reasons we say we are not the same" (pregnant woman, FGD 4).

Second, participants explained that while "we all get the social grant", mothers' capability to use the grant to enhance nutrition differed when "there's just 3 of them in a household, while others there are 10 of them in a household" (mother of young infant, FGD 3). Lack of affordable housing inevitably resulted in overcrowding. With many mouths to feed social grants and food aid did not last long. It seemed unrealistic that the Child Support Grant or food aid would be safeguarded only for the mother and child because "that grant has to stretch for everyone, even for the unborn child" (CBO 4, woman, social worker). Mothers' and children's nutrition needs based on clinic recommendations could not be met. Participants explained that as a mother, "after giving birth you would still stay with other people, so you won't be able to buy something that accommodates only you and say: I have a child and therefore have to put this type of food aside, that I have to eat" (mother of young infant, FGD 2).

Furthermore, some mothers lived in a household without stable electricity, refrigeration, and inadequate storage space. This hindered mothers' capability to purchase fresh produce because "if you buy something [fresh produce] at Bara [locally], it won't last you a long time [...] since we don't have electricity, food gets rotten" (mother of young infant, FGD 6). Even the storage of perishable food was a problem. This restricted mothers' participation in stokvels where food was usually purchased in bulk at the end of the year. Participants recognised both the potential benefits of stokvels as well as their own inability to participate: "some can buy a lot of the grocery that is not so perishable, and they will last for six months. The problem is we live in small houses and have storage issues" (mother of young infant, FGD 6).

### Access to civic services, welfare and the labour market

Research participants expressed that some variation in mothers' capability to benefit from services could be avoided if barriers in access to civic services, welfare and labour market were removed. First, participants discussed access in terms of time and transportation costs. Located outside of the township, access to employment opportunities and government departments required mothers to spend a significant amount of their time on travelling, and between ZAR 20–100 (US$1–6) on transportation fees. As a CBO participant explained, "Let's say the ZAR 350 [USD 22] SASSA [South African Social Security Agency] money is in. I'm hungry now, I know the money is there, but I don't have transport to go get the money. That already affects nutrition because I don't have access" (CBO 1, women, service provider). Similarly

valuable yet often inaccessible services were the Department of Social Development's food relief and social work services, the Department of Human Settlements for subsidised housing and the Department of Home Affairs for civic services. The latter was of particular concern since without correct documentation (identification document, birth certificates) mothers and children could not qualify for social grants and food parcels. Furthermore, those without identity documents were also unable to obtain a matriculation certificate on secondary school completion; this further restricted mothers' access to the labour market, perpetuating a low-skill/low-income trap for some mothers. Besides constraints in physical access, mothers differed by their level of life skills. This manifested in low levels of internet literacy, not knowing how to fill out necessary documents for government services, where to look for employment opportunities, and how to write job applications and CVs. These processes were also not without costs (e.g., printing, internet data), and required sometechnical skills that further excluded some low-skill/low-income mothers.

### Social support
#### Social cohesion and connectedness

Social groups were seen as a potential buffer against food insecurity, stress, isolation and lack of belonging in the community. A FGD participant described how one could benefit from being part of a group: "at our church we contribute something every month—being it tinned-food or anything else [...] we take the groceries to those needy families we know of. We also sit down as women and talk about the challenges we have" (mother of young infant, FGD 3). Mothers who were socially connected were seen more resourceful and better able to access information about potential opportunities (including jobs, income generating groups, entrepreneurial advice), and circulated such information among these social networks. A CBO participant noted that "some of the resources are in churches, some are in other people just saying, 'I can show you how to write a CV' [...] that empowers you to then be able to look for a job better" (CBO 10, women, social worker). Nevertheless, FGD participants reported that lack of trust and feelings of exclusion limited some mothers' capability to access and use certain resources. FGD participants saw poverty alleviation projects and food aid as important services for nutrition but of lower value due to their feelings of mistrust. Benefits of these services were seen contingent on having close ties with service providers.

Age was also believed to contribute to variation in mothers' capability to benefit from employment opportunities, and poverty alleviation projects. Both CBO and FGD participants saw older mothers (35>) being able to benefit less from economic opportunities, while younger FGD participants felt excluded from poverty alleviation projects. A participant illustrated this point by sharing her experience of trying to join a gardening group: "The granny I asked if I can't just plant even if it is a few

onions on the side, said that it was for the elderly. She said I could—because I was still young—go find a job in an office" *(mother of young infant, FGD 6)*. When opportunities were filled or groups were full, FGD participants saw this as the manifestation of "corruption" in the community. FGD participants' observations on this resonated with that of CBO representatives. However, perspectives differed about why this was the case. CBOs emphasised the scarcity of poverty alleviation projects in the community; hence, opportunities filled up fast. Further, CBOs' eligibility criteria for beneficiaries could lead some mothers to feel excluded and believe that CBOs "have their own people" *(mother of young infant, FGD 2)*.

### Psychosocial support

The degree to which mothers benefitted from clinic-based services depended on their perception of the stigma associated with seeking help. Engagement with social workers at clinics was associated with fear of judgement by others in the community. The negative attitude of healthcare workers towards young mothers at antenatal and postnatal visits often left mothers feeling devalued and unaccepted. Low self-esteem was also associated with young women's position in their households where one's acceptance and value were contingent on financial contribution. Despite the theoretical advantages of social support, large family networks did not seem to translate into benefit for mothers. Participants described displays of negative attitudes of family members when mothers, especially those still at school, had babies. Additional expenses related to the newborn were often perceived as cumbersome for low-income families. Resulting inadequate financial and childcare support for these young mothers were seen as significant contributors to school dropouts, and a barrier to participate in skills training or income generating activities. Furthermore, those without adequate psychosocial support were more likely to engage in risky coping strategies including substance abuse, gambling, debt and remaining in abusive relationships. The benefit of food vouchers and social grants for mothers' and children's nutritional status was limited when these scarce resources were spent on harmful habits or when abusive relationships took away mothers' decision-making power.

### Access to affordable healthier foods

Participants confirmed that most mothers in the community were aware and willing to follow clinic-based nutrition advice and wished to spend social grants in a more 'suitable' way for nutrition. However, their capabilities to execute their decisions were hindered by lack of finances. One FGD participant explained, "even though we can be told the same advice—our pockets sizes are not the same. I don't have the money to buy the things I was advised to buy" *(pregnant woman, FGD 4)*. Foods recommended at the clinics were perceived to be expensive and being able to afford them was the exception rather than the norm. Access to healthier food was restricted to a few days in a month when mothers had just received the Child Support

Grant: "it's only when you get the social grant money that you can buy that, then you have at least a banana and yoghurt. During the month, you don't have" *(mother of young infant, FGD 6)*. Despite fruits and vegetables being available locally at a low cost, FGD participants explained that mothers "have to go to Woolworths [high-end supermarket], because veggies here at the township are not fresh" *(pregnant woman, FGD 1)*. However, mothers' access to supermarkets depended on their ability to pay for transportation. When the opportunity cost of accessing healthier foods was too high, some mothers had to compromise the quality of food over its quantity. FGD 2 participant explained that "nutritional food is more expensive. And then its quantity becomes little. When you look at the families there are many of us. […] How long will it last. So, when you buy, you weigh your options. Even with the Rama [margarine]. You buy a cheap one, which does not have the right nutrition" *(mother of young infant)*. Mothers with limited access to supermarkets relied on local shops that were described as sources of inexpensive but unhealthy foods. "They sell kotas [street food]—There's nowhere they sell sandwiches, salads" *(pregnant woman, FGD 4)*.

### DISCUSSION

Despite free primary healthcare services and a well-developed social protection system for mothers and children, there are significant inequalities in nutrition outcomes across South Africa. Much attention has been paid to addressing nutrition inequalities through increasing universal health coverage,[32 33] and through removing access barriers to services.[34 35] Less consideration has been given to what determines the realisation of benefits, once services are available and accessible. We developed a framework anchored in ecological and capability approaches and defined social needs as the requisites that can magnify (if unmet) or reduce (if met) variation in the degree to which individuals can benefit from existing services. We applied this framework to identify social needs that determine maternal and child nutrition through engaging pregnant women, mothers, and CBOs in an urban township community.

From a policy perspective, this is relevant, because the extent to which existing services could enhance maternal and child nutrition will be moderated by mothers' capability to benefit from these services. To this end, our study fills several gaps in the literature including what drives persisting nutrition inequalities and how we design nutrition policies. Recent commentaries have highlighted the limitations of current micro level and macro level approaches in understanding health inequalities.[36 37] Academics have called for new narratives that take account of "the variation within groups as a way to better understand the dynamics of inequality" (Lundberg 2020, 36). Our social needs approach locates these variations at the meso level of the ecological model and highlights how the meso level forms a link between the macro level and micro level in producing patterns of nutrition inequalities

(figure 1). This is key, because to date, studies have focused on the absolute effect of determinants and services (macro level) on nutrition (micro level). Furthermore, efforts to increase access and utilisation of services to improve nutrition in low- and middle-income countries have focused mostly on health sector interventions.[38 39] The social needs approach extends this focus to multiple sectors by considering all types of services that are perceived as valuable for mothers to ensure optimal nutrition.

We found that in an urban township, the capability of mothers to benefit from eight valued services to keep themselves and their children well-nourished depended on their ability to meet six social needs. Of these, some needs — for example, personal income stability — have been identified before. Previous studies have recognised financial constraints as a key barrier for some mothers to access healthier foods in Soweto.[40 41] These have frequently been linked to lack of financial support from fathers, and consequent dependency on social grants.[40–42] Previous analyses have revolved around the impact of financial constraints on food related perceptions, purchasing and dietary behaviours. The impact of financial barriers on individuals' interactions with available services that they see as valuable for nutrition has been less of a focus. The value of the social needs approach in the present study lies not only in establishing that lack of personal finances limited mothers' nutrition, but in identifying which were those services that—when mothers were deprived of—limit optimal nutrition. In our study, nutrition was compromised when mothers without an additional income source (other than social grants) were deprived from the benefits of poverty alleviation projects, crèches with access to nutritious meals, and clinic-based nutrition advice. Low income also limited mothers' ability to attend skills training workshops, thereby depriving them from the benefits of economic progress and subsequent improved nutrition.

Through the social needs lens, this study extended previously described associations between suboptimal nutrition and unemployment[40] by identifying reasons that keep mothers unemployed, and what it would take to enhance their capability to benefit from available job opportunities. The government's primary strategy to unemployment includes job creation.[43] Notwithstanding the need for more jobs at the macro level, we showed that mothers in this study needed more immediate and short-term support including technological skills and internet literacy, writing CVs, preparing for job interviews, and incidental expenses linked to job applications. Like mothers in this study, Maharaj and Dunn (2022) found that young Black African women in a urban township in Durban, South Africa, could not go out and search for employment because they did not have the means to do so.[44]

We also showed that the same unmet needs constrained the ability of mothers to benefit from various government services. These included civic services (birth certificates, identity documents) and welfare (social grants, food parcels, social housing). With regards to the latter, unmet need for appropriate and affordable housing was seen to be responsible for a large share of the variation in benefit from social grants and food aid. Mothers' ability to direct service benefit towards children's nutrition needs was diminished when social grants and food had to cater for many in overcrowded households and cover all household expenses. Nutrition as recommended at the clinic was also compromised when mothers' needs for stable electricity, refrigeration and storage space in the households were unmet. While the link between nutrition and poor living conditions has been discussed in previous studies of the study area, these emphasised the role of poor sanitation and hygiene in malnutrition.[40] Here, we showed how government resources (social grants and food aid) may lose their effectiveness if needs for affordable and appropriate housing are unmet.

A social need linked to finances, but less well documented, was financial planning skills. Nutrition was compromised when mothers lacked the financial acumen to manage scarce resources including social grants and food vouchers. Well-planned finances on the other hand were seen as a protective factor against running out of resources by the end of the month. Furthermore, financially literate mothers were seen to use social grants and food vouchers in a way that was better for nutrition. We identified that meeting mothers' need for financial planning skills could reduce avoidable variation in benefiting from stokvels, poverty alleviation projects and skills development trainings.

A realm of studies has identified social support as imperative to buffer against stress during and after pregnancy and promote healthier mother and child nutrition.[45 46] Social support, however, has not been readily available to South African mothers. Like some participants in this study, researchers have associated this unmet need with the absence of fathers, negative provider–patient interaction at clinics,[46–49] and stigma attached to being pregnant at a young age.[42 50] This study extends the literature on stigma by documenting mothers' perceptions of stigma associated with seeking assistance to cope with financial and nutrition difficulties from social workers and/or informal networks. This is key to the discussion around nutrition inequalities because evidence from high-income countries has shown how variation in health benefits might be partially explained by the stigmatising effect of services.[51] Furthermore, we showed how mistrust and feelings of exclusion from available services limited mothers' ability to forge support networks that could help to deal with their issues. Participants in this study found food aid and poverty alleviation projects to be important for nutrition but of lower value because they mistrusted the service provider and felt excluded from the projects. Mistrust of services in this way also contributes to variation in mothers' capacity to benefit from services and produces nutrition inequalities as a consequence.

## Strengths and limitation

This is the first qualitative study in South Africa to understand the variation of nutrition outcomes of mothers and children through a social needs approach. A key strength of the study design was engaging with multiple stakeholder groups, which enhanced both the validity and comprehensiveness of the data. Nevertheless, stakeholders' perspectives were not representative of all South African mothers. Participants represented the experiences of lower income mothers in a resource-constrained urban township. Social needs however may take different forms at different locations (e.g., rural areas). For this study purposes, it was important however to demonstrate that social needs are context dependent, and researchers and policy-makers should challenge the way current services are designed and targeted.

## Conclusion

Mothers and pregnant women differ in their abilities to benefit from available healthcare services and social protection services. This has implications for them and for their children's nutritional health. Investigating the variation in mothers' ability to benefit from services might help understand why nutrition inequalities persist despite well-developed policies. A social needs approach can help identify what additional support mothers require to turn available services into nutrition benefits. Multisectoral strategies are needed to address social needs and to counterbalance narrow clinic-based nutrition services and high-level population instruments. A social needs approach could complement and reduce variation in the effect of existing services that contribute to inequalities in mother and child nutrition outcomes.

**Author affiliations**
[1]SAMRC/Centre for Health Economics and Decision Science—PRICELESS SA, School of Public Health, Faculty of Health Sciences, University of Witwatersrand, Johannesburg, South Africa
[2]School of Economics and Finance, University of the Witwatersrand, Johannesburg, South Africa
[3]School of Health Sciences, Faculty of Environmental and Life Sciences, University of Southampton, Southampton, UK
[4]MRC Lifecourse Epidemiology Centre, University of Southampton, Faculty of Medicine, Southampton, UK
[5]School of Public Health, Faculty of Health Sciences, University of the Witwatersrand, Johannesburg, South Africa

**Acknowledgements** The authors wish to thank Sandra Ntebe for her contribution to the fieldwork and to the participants for taking part in this study.

**Contributors** AE conceptualised the study with input from CD, KH, MB and NJC. AE and CD developed the study framework, and AE and NJC designed the data collection tools. AE collected and analysed the data. NJC guided data analysis and interpretation, with input from all coauthors. AE drafted the manuscript, and all coauthors edited the manuscript and approved the final version. AE is responsible for the overall content as guarantor.

**Funding** This research was funded by the National Institute for Health Research (NIHR) (17\63\154) using UK aid from the UK Government to support global health research (https://www.nihr.ac.uk/). The views expressed in this publication are those of the authors and not necessarily those of the NIHR or the UK Department of Health and Social Care. The funders had no role in study design, data collection and analysis, decision to publish, or preparation of the manuscript. AE, CD, and KH are supported by the SAMRC/Wits Centre for Health Economics and Decision Science—PRICELESS SA (grant number 23108). AE has received funding from the Faculty of Health Sciences, University of the Witwatersrand, Johannesburg, South Africa.

**Competing interests** None declared.

**Patient and public involvement** Patients and/or the public were involved in the design, or conduct, or reporting, or dissemination plans of this research. Refer to the Methods section for further details.

**Patient consent for publication** Not applicable.

**Ethics approval** The study was conducted according to the principles of the Declaration of Helsinki, and all procedures involving research study participants were approved by the University of the Witwatersrand Human Research Ethics Committee (M210718) and the Research Committee of Johannesburg Health District (GP_202110_002). All participants were informed that participation in the study was voluntary and reassured about the confidentiality of the data collected. Written informed consent was gathered from each participant.

**Provenance and peer review** Not commissioned; externally peer reviewed.

**Data availability statement** Data are available upon reasonable request. The dataset generated and analysed during the study is not publicly available due to limitations of ethical approval regarding participant confidentiality.

**ORCID iD**
Agnes Erzse http://orcid.org/0000-0001-9303-9323

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
