## [Reviewer comments · BMJ Open]

ARTICLE DETAILS

TITLE (PROVISIONAL)	A qualitative exploration of the constraints on mothers' and pregnant women's ability to turn available services into nutrition benefits in a low resource urban setting, South Africa
AUTHORS	Erzse, Agnes; Desmond, Chris; Hofman, Karen; Barker, Mary; Christofides, Nicola

VERSION 1 – REVIEW

REVIEWER	Akter, Fahmida BRAC University James P Grant School of Public Health
REVIEW RETURNED	18-Jun-2023

GENERAL COMMENTS	Overall comments: The authors explored one of the timely and evolving issues which will enrich the knowledge in the area of child and maternal nutrition service utilization in low resource urban context like Soweto in South Africa. The findings of this study are important to explain the sources of variability among pregnant women and mothers to convert the available nutrition services to actual nutrition benefits. Thank you for giving me the opportunity to review this important article. This is a very well written manuscript with sound theoretical interpretation. I have identified several issues authors need to address as follows: Specific comments: Title: The authors could revise the current title of the manuscript to make it more reflective, explicit, and aligned with the manuscript. Particularly the phrase 'capacity to benefit from the service environment to enhance nutrition' seems a bit deviated to me (the term 'service environment' and 'enhance nutrition'). Instead, authors could make it more direct (as stated in several sections in the main text of the manuscript). Abstract: The authors did not fully follow the prescribed structure for abstract of this journal (https://bmjopen.bmj.com/pages/authors) [objectives, design, setting, participants...conclusion]. Introduction: • When introducing an acronym for the first time, mention its elaboration (such as ZAR). Methods: Sampling • 'Fifteen CBOs were contacted, and 10 were included in the final sample'. Please mention, how many eligible CBOs were identified at first, then how many was contracted (and why), and then how 10
--

	were selected.  • 'purposely sampled pregnant women attending antenatal care and mothers attending postnatal care' Please specify 'mothers' (such as with at least one under five years old children, or mother with 0-23 months old child or any mothers etc.) • 'at five primary health care facilities in Soweto' please mention, how these five PHC facilities were selected. • Please revise the title of Appendix-1 and add few more information on the FGD participants in the table. Methods: Data collection  • 'An interview and a FGD guide were developed separately to assist data collection.' Please add the interview and FGD guides as supplementary materials. • 'The feasibility of our data collection tools was verified through a pilot interview and FGD.' Where did you do the pilot? Within or outside the study? Please mention. • 'Written consent was taken from all participants' Please mention the term 'written' in 'Ethics statements' section too. Methods: Data management and analysis  • 'Once inductive and deductive codes were developed, codes were organized into broader categories based on thematic similarities' please add a chart or table as supplementary material with all codes within categories and all categories within theme for reader's better understanding. • 'Reporting of the findings adheres to COREQ guidelines' please add the filled-up checklist as a supplementary material. Results: Characteristics of CBO participants  • The author could add 'years of experience in this sector' as one of the characteristics (if data is available). • Table 2: Source of income: Please add few examples of 'others in the household'. Results: Social needs of mothers  • 'The analysis identified six social needs' Has the author ranked the needs in any way? Or is it arranged in accordance with the most cited needs? Please mention in the manuscript. • 'that enabled or disabled mothers from benefitting from eight different services' How did the authors come up with the list of these eight services? How was the question asked about the services? Was it list based? Or open? • 'These services were recognized as valuable for nutrition' by who? FGD participants? CBOs? Or both? • Please make it clear in the entire result section that findings are from provider side (CBOs) or from recipient side (FGD participant) or supported by both? Are there any findings where these two sides reported differently? Results: Financial planning for nutrition  • After each quotation, please add a bit detail about the respondent keeping the full confidentiality. For example, if the quotation is by CBO: (CBO 1, sex (male/female), designation). If the quotation is from a FGD participant: (pregnant women/mother of x children, FGD x). Discussion:  • The authors could add few references of findings from similar context along with the references for same study areas.
--	---

	Ethics statement:  • Mention that written consent was taken. References:  • Please check reference no 17 and 19. Also check if all references are in same format or not. Supplementary material reporting  • For the readers' better understanding, the authors should include interview and FGD guides as supplemental materials. • Please add a chart or table as supplementary material with all codes within categories and all categories within themes for reader's better understanding. • The authors should include COREQ checklist where they report the page number in the manuscript for each of the items listed in this checklist. • Appendix 1:  o The title needs to be self-explanatory. o Add the unit of age (i.e., year).
--	---

VERSION 1 – AUTHOR RESPONSE

Reviewer: 1 Dr. Fahmida Akter, BRAC University James P Grant School of Public Health	Authors' response
Overall comments: The authors explored one of the timely and evolving issues which will enrich the knowledge in the area of child and maternal nutrition service utilization in low resource urban context like Soweto in South Africa. The findings of this study are important to explain the sources of variability among pregnant women and mothers to convert the available nutrition services to actual nutrition benefits. Thank you for giving me the opportunity to review this important article. This is a very well written manuscript with sound theoretical interpretation. I have identified several issues authors need to address as follows:	The authors thank the Reviewer for the positive and constructive feedback. Please note that page number references in our responses below are based on the "Main Document – marked copy".
Title: 1. The authors could revise the current title of the manuscript to make it more reflective, explicit, and aligned with the manuscript. Particularly the phrase 'capacity to benefit from the service environment to enhance nutrition' seems a bit deviated to me (the term 'service environment' and 'enhance nutrition'). Instead, authors could make it more direct (as stated in several sections in the main text of the manuscript).	Thank you for the suggestion. The title has been revised accordingly and reads as follows: "A qualitative exploration of the constraints on mothers' and pregnant women's ability to turn available services into nutrition benefits in a low resource urban setting, South Africa".
Abstract:	Thank you for pointing this out. The abstract

2. The authors did not fully follow the prescribed structure for abstract of this journal (https://bmjopen.bmj.com/pages/authors) [objectives, design, setting, participants...conclusion].	has been revised accordingly.
Introduction: 3. When introducing an acronym for the first time, mention its elaboration (such as ZAR).	Thank you, the text has been modified as follows: (South African Rand [ZAR] 450 or US\$26/month as of 2023) (page 3)
Methods: Sampling 4. 'Fifteen CBOs were contacted, and 10 were included in the final sample'. Please mention, how many eligible CBOs were identified at first, then how many was contracted (and why), and then how 10 were selected.	4. Thank you. This has now been clarified by revising the sentence as follows: "Out of the fifteen CBOs that were identified and contacted, 10 responded to the invitation to participate, thus comprising our final sample." (page 6)
5. 'purposively sampled pregnant women attending antenatal care and mothers attending postnatal care' Please specify 'mothers' (such as with at least one under five years old children, or mother with 0-23 months old child or any mothers etc.)	5. Thank you. Clarification and revision have been made throughout the manuscript. The revised wording is the following: "mothers with infants under one year of age" (e.g., page 5, 6, 7)
6. 'at five primary health care facilities in Soweto' please mention, how these five PHC facilities were selected.	6. Clinics were purposively sampled with consideration to ensure representation from various suburbs. Revisions have been made accordingly on page 6: "Clinics were purposefully selected to ensure representation from various suburbs, allowing for a more comprehensive collection of perspectives and experiences."
7. Please revise the title of Appendix-1 and add few more information on the FGD participants in the table.	7. Thank you for the suggestion. Please note that Appendix 1 has been renamed to supplemental file 1. As suggested by the Reviewer, the title has been revised as the following: "Composition of the seven focus groups held with pregnant women and mothers with infants under one year of age" Furthermore, three columns have been added, outlining the following information:  - Participants with a household income <=3000 ZAR/month (n) - Participants with >=2 children (n) - Mean number of household members (SD)
Methods: Data collection 8. 'An interview and a FGD guide were developed separately to assist data collection.' Please add the interview and FGD guides as supplementary materials.	8. FGD and interview guides have been added and referred to as supplemental file 2 and 3. (page 6)

9. 'The feasibility of our data collection tools was verified through a pilot interview and FGD.' Where did you do the pilot? Within or outside the study? Please mention.	9. Thank you for raising this important point. We have modified the text in the following way: "A pilot to test the data collection methods was undertaken with participants who met the eligibility criteria. The pilot was conducted in the study setting." (page 6)
10. 'Written consent was taken from all participants' Please mention the term 'written' in 'Ethics statements' section too.	10. Thank you. We have corrected the wording accordingly.
Methods: Data management and analysis 11. Once inductive and deductive codes were developed, codes were organized into broader categories based on thematic similarities' please add a chart or table as supplementary material with all codes within categories and all categories within theme for reader's better understanding.	11. A code structure has been added – please refer to supplemental file 4.
12. 'Reporting of the findings adheres to COREQ guidelines' please add the filled-up checklist as a supplementary material.	12. The checklist with indication of the page number where information can be found has been provided in supplemental file 5. Please note that we have added additional information to the manuscript on two occasions so the reporting of our methods are in line with the guidelines. These include the following sentences on page 7: "The interviews were conducted in English by the first author, who had experience in qualitative interviewing and working with communities in Soweto, conducted the interviews in English, either in person at the CBO or via telephone." "Transcripts were analysed progressively to determine the point at which saturation was achieved."
Results: Characteristics of CBO participants 13. The author could add 'years of experience in this sector' as one of the characteristics (if data is available).	13. Data is available and the following sentence has been added to the paragraph: "Thirteen of the participants indicated that they had been employed at their respective CBOs for over 4 years." (page 8)
14. Table 2: Source of income: Please add few examples of 'others in the household'.	14. The following examples have been added to the bottom of Table 2: "*Infants'/children's father, grandparents, other cohabiting adults"
Results: Social needs of mothers 15. The analysis identified six social needs' Has the author ranked the needs in any way? Or is it arranged in accordance with the most cited needs? Please mention in the manuscript.	15. Thank you. The reporting of the social needs (seen as most pressing based on the quantity of discussion), does not follow a priority rank order. This has now been clarified in the manuscript: "The following section reports on each of these needs without implying a priority rank order, as they link to multiple sources of variation that enabled or disabled mothers from benefitting from eight different services." (page 9)

	Furthermore, we have removed the numbering of social needs in table 3, to avoid implying a ranking among the needs.
16. 'that enabled or disabled mothers from benefitting from eight different services' How did the authors come up with the list of these eight services? How was the question asked about the services? Was it list based? Or open?	16. Thank you for the important question. The list of eight services was based on participant's responses, not the authors. This has been implied in the original submission on page 7, where we explained that participants were asked "about their perceptions of the challenges that affected maternal and child nutrition in the community, as well as the available resources, or services that they knew of and might help addressing these challenges". The newly included interview and FGD guides found supplemental files 2 and 3 offer readers additional insight into the kinds of questions and prompts employed to gather information about services that participants deem important.
17. 'These services were recognized as valuable for nutrition' by who? FGD participants? CBOs? Or both?	17. We have clarified this in the text indicated that we talk about both groups of participants. See page 7: "...benefit from eight different services that both CBOs and mothers were identified as valuable for optimal maternal and child nutrition..." See page 9: "These services were recognized by CBOs and mothers as..."
18. Please make it clear in the entire result section that findings are from provider side (CBOs) or from recipient side (FGD participant) or supported by both? Are there any findings where these two sides reported differently?	18. Thank you for the important suggestion. We have added the following sentence to indicate the approach we took when reporting on the results: "Where results are similar for CBOs and mothers, these are presented without participant specification. When results differ, these are noted and discussed." (Page 7-8) Example of different perspectives in the original submission include: "However, perspectives differed about why this was the case. CBOs emphasized the scarcity of poverty alleviation projects in the community, hence opportunities filled up fast." (Page 13)
Results: Financial planning for nutrition 19. After each quotation, please add a bit detail about the respondent keeping the full confidentiality. For example, if the quotation is by CBO: (CBO 1, sex (male/female), designation). If the quotation is from a FGD participant: (pregnant women/mother of x	19. Thank you for the suggestion. We have made the recommended specification for each quote in the manuscript.

children, FGD x).	
Discussion: 20. The authors could add few references of findings from similar context along with the references for same study areas.	20. Thank you for the suggestion. In the first paragraph of the discussion, we have replaced the references citing papers with global or regional focus with South Africa specific papers, and 2 papers from the study area. In the rest of the discussion, references 38-40, 44 (in the original submission) were from the study area, Soweto.
Ethics statement: 21. Mention that written consent was taken.	A specific "Ethics statement" section has been added with the recommended wording. (Page 16-17)
References: 22. Please check reference no 17 and 19. Also check if all references are in same format or not.	22. Thank you for pointing this out. We apologize for the mistake. The intended reference here was Bhutta et al that outlines nutrition sensitive and specific strategies. The duplication of 17 and 19 has been addressed by replacing the Black et al reference with Bhutta et al. Formatting has been checked and corrected.
Supplementary material reporting 23. For the readers' better understanding, the authors should include interview and FGD guides as supplemental materials.	23. Interview and FGD guides have been added, see supplemental file 2 and 3.
24. Please add a chart or table as supplementary material with all codes within categories and all categories within themes for reader's better understanding.	24. A table with high level code structure has been added, see supplemental file 4.
25. The authors should include COREQ checklist where they report the page number in the manuscript for each of the items listed in this checklist.	25. A COREQ checklist has been provided with page numbers indicating where in the manuscript are each of the items listed in this checklist appear, see supplemental file 5. (Please note that page numbers might require revisions once the tracked changes have been accepted).
Appendix 1: 26. The title needs to be self-explanatory.	26. Appendix 1 has been renamed as Supplemental file 1. The title has been revised as follows: "Total number of focus groups held by sociodemographic characteristics of participants"
27. Add the unit of age (i.e., year).	27. Thank you. The unit has been added as suggested.

VERSION 2 – REVIEW

REVIEWER	Akter, Fahmida BRAC University James P Grant School of Public Health
REVIEW RETURNED	17-Oct-2023
GENERAL COMMENTS	Overall comments: The authors addressed all the points previously mentioned. Thank you for the revised version. I have no further comment.